# Context-Sensitive Semantic Reasoning in Large Language Models

**Tyler Giallanza & Declan Campbell**
Department of Psychology & Princeton Neuroscience Institute
Princeton University
Princeton, NJ 08544, USA
`{tylerg, idcampbell}@princeton.edu`

## Abstract

The development of large language models (LLMs) holds promise for increasing the scale and breadth of experiments probing human cognition. LLMs will be useful for studying the human mind to the extent that their behaviors and their representations are aligned with humans. Here we test this alignment by measuring the degree to which LLMs reproduce the context-sensitivity demonstrated by humans in semantic reasoning tasks. We show in two simulations that, like humans, the behavior of leading LLMs is sensitive to both local context and task context, reasoning about the same item differently when it is presented in different contexts or tasks. However, the representations derived from LLM text embedding models do not exhibit the same degree of context sensitivity. These results suggest that LLMs may provide useful models of context-dependent human behavior, but cognitive scientists should be cautious when assuming that embeddings reflect the same context sensitivity.

## 1 Introduction

The study of how humans learn about, represent, and reason about objects and their properties has long been a fundamental topic in cognitive psychology (Collins & Loftus, 1975; Murphy, 2004; Rips et al., 2012). Despite decades of research into these questions, however, most models of conceptual knowledge operate only on small datasets and fail to capture the full richness of human semantic cognition (Bhatia & Richie, 2022). Recent work has proposed using advances in natural language processing, particularly large language models (LLMs), to scale models of semantic cognition to larger datasets (Bhatia, 2023; Bhatia & Richie, 2022). These approaches generally combine knowledge representations learned by LLMs with theories of learning, decision making, and reasoning developed in cognitive science. Leveraging LLM representations in this way scales the testing of psychological theories to larger datasets and domains in which collecting human behavioral data would be prohibitively costly (e.g., pairwise similarity; Iordan et al., 2022; Marjieh et al., 2022).

Using LLM representations to study human cognition relies on the assumption that LLM and human representations are aligned. This appears to be largely the case, particularly when LLMs are fine-tuned on human data: Recent work has shown that these models provide a strong fit to human semantic similarity judgments, inductive reasoning, feature verification, and typicality judgments (Bhatia & Richie, 2022; Bhatia, 2023; Suresh et al., 2023). However, these tests of LLM-human alignment have largely focused on context-independent forms of representation. In contrast, human knowledge representations are dynamic and context-sensitive, which has been shown at both the behavioral (Giallanza et al., 2023) and neural (Cukur et al., 2013) level.

To address this gap, in this article we measure how well LLMs capture two forms of context sensitivity observed in human behavior and knowledge representations, which we refer to as *local context sensitivity* and *task context sensitivity*. First, humans are sensitive to objects that are present in the environment but not directly related to the task at hand (local context sensitivity; Tversky & Gati, 1978). For example, Sjöberg & Thorslund (1979) showed that the similarities people report between string instruments (e.g., the similarities between banjos, violins, harps, and electric guitars) were increased when a wind instrument (e.g., a clarinet) was added to the set. Second, humans rely on

different representations for the same object when tasks require attending to different features (task context sensitivity; Cukur et al., 2013; Giallanza et al., 2023; Iordan et al., 2022). This finding holds both neurally (Cukur et al., 2013), as attending to either humans or vehicles in a visual search task alters neural representations to selectively represent objects from the relevant category, and behaviorally (Giallanza et al., 2023), as asking humans to judge the similarity between objects *in terms of size* requires different representations than standard, unconstrained similarity judgments. In the remainder of this article, we test how well LLM behavior and LLM representations account for these context effects by simulating tasks from the cognitive science literature. The results demonstrate that LLM behavior shows context sensitivity, while LLM embedding representations are substantially less context sensitive.

## 2 LOCAL CONTEXT SENSITIVITY

Extensive work in cognitive science has demonstrated that human behavior is sensitive to irrelevant items present in the environment at or around the time a task is being performed (Sjöberg & Thorslund, 1979; Tversky & Gati, 1978; Tversky & Kahneman, 1974). In the domain of semantic decision making, people produce different similarity ratings for the same pair of objects depending on the broader set of objects used in the experiment. In particular, multiple studies have found that the reported similarity between items that share a common feature increases when an additional item that does not share that feature is introduced (e.g., banjos and violins appear more similar when the previous trial involves a clarinet; Sjöberg & Thorslund, 1979; Tversky & Gati, 1978).

Tversky & Gati (1978) measured this using countries from different regions. 120 participants judged the similarity between pairs of countries in one of two conditions (Figure 1a). In the *homogeneous* condition, each participant saw a list of countries from the *same* region (either Europe or North/South America, randomly assigned to each participant). They were presented with 8 different pairs of countries from that region and asked to rate the similarity of the two countries in each pair on a scale of 1 to 20. In the *heterogeneous* condition, each participant instead saw a mix of countries from the two regions. They were presented with 4 country pairs from the European region and 4 from the American region. Tversky & Gati (1978) found that the similarity ratings differed across these two conditions. People tended to give a higher similarity score for the same country pair (e.g., Slovenia - Austria) when that pair was presented in the heterogeneous condition than when it was presented in the homogeneous condition. Tversky & Gati (1978) posited that this is because the heterogeneous condition, which contains both European and American countries, increases the salience of geography, biasing people to find countries within the same region (e.g., Slovenia and Austria) more similar to one another.

### 2.1 RESULTS

We simulated the Tversky & Gati (1978) experiment using both GPT-3.5 and GPT-4. We prompted each model using the same instructions that were provided to human participants (i.e., "The following is a list of 8 pairs of countries. Please indicate, for each of the 8 country pairs, how similar the two countries are to one another on a scale of 1 to 20. Pair 1: Slovenia - Austria. Pair 2: Bulgaria - Greece. etc."), collecting LLM similarity ratings across the homogeneous and heterogeneous conditions. Figure 1c shows the similarity scores provided by humans and the LLMs for each of the 16 country pairs. The direction of the arrows represents the change in similarity score from the homogeneous condition to the heterogeneous condition. For example, the average participant rated the similarity for the pair "Slovenia-Austria" as $8.47$ in the homogeneous condition and $9.86$ in the heterogeneous condition. Because the rating is larger in the heterogeneous condition, the arrow points to the right, indicating an effect in the predicted direction. We measured the significance of the effects for humans and each of the LLMs using a one-tailed paired t-test that measures if similarity ratings for the same country pair are significantly greater in the heterogeneous condition than in the homogeneous condition. Both human ratings and GPT-4 ratings showed a significant effect in the predicted direction (human: $t = 2.11$, $p = .026$; GPT-4: $t = 4.3$, $p = .003$), but GPT-3.5 ratings did not ($t = -0.083, p = .53$).

We next investigated why GPT-3.5 fails to demonstrate a significant effect. One possibility is that GPT-3.5 is inherently less sensitive to local context than is GPT-4, while another possibility is that GPT-3.5 is less capable of producing human-like similarity ratings. We tested this by correlating

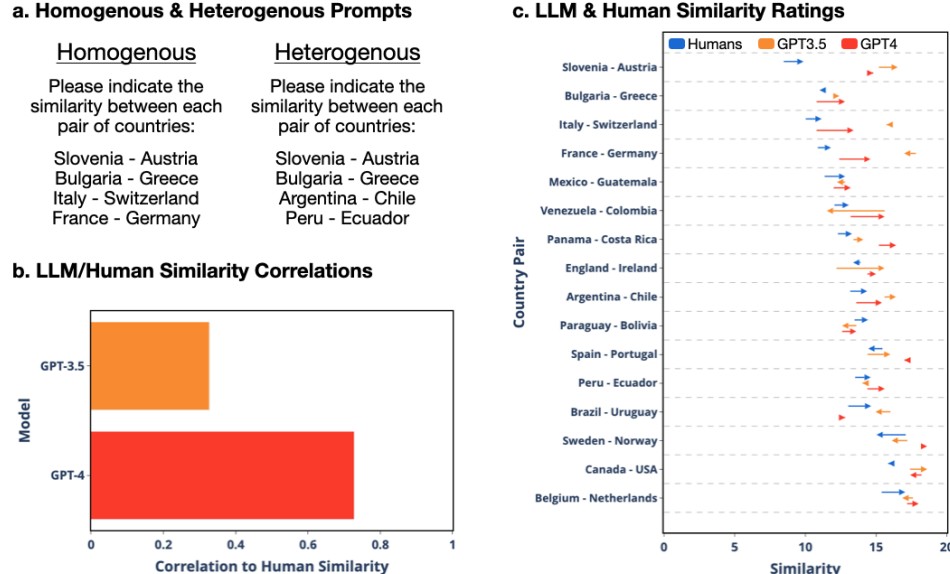

Figure 1: **Local context effects in humans & LLMs.** a) The prompt in the homogeneous condition contains country pairs from the same region, while in the heterogeneous condition half of the pairs contain European countries and the other half contain American countries. b) GPT-4's similarity ratings correlate strongly with human similarity ratings, but GPT-3.5's ratings do not. c) Similarity scores provided by each model for each country pair across the homogeneous and heterogeneous conditions. Arrows indicate the change in similarity score from the homogeneous to the heterogeneous condition, where arrows pointing to the right indicate an effect in the predicted direction.

the similarity scores provided by the LLMs to the similarity scores provided by humans (averaged across the homogeneous and heterogeneous conditions; Figure 1b). The results show that GPT-4's similarity ratings correlate strongly with human ratings (Pearson $r = 0.73$), while GPT-3.5's ratings provide a weak fit ($r = 0.33$). This suggests that the inability for GPT-3.5 to reproduce human-like biases towards context sensitivity in this task may simply result from its inability to judge the similarity between countries in a human-like way.

## 3 TASK CONTEXT SENSITIVITY

A salient feature of human semantic cognition is our ability to respond to the same item in different ways depending on the task that needs to be performed (Ralph et al., 2017). For example, we can view a basketball as a type of ball when at the gym, as an orange sphere when painting, or as a medium-sized, light object when packing for a trip. Both neural and behavioral evidence suggests that this flexibility in behavior arises from flexibility in representation: in any given context, humans selectively represent the context-relevant features or properties of the item while suppressing context-irrelevant ones (Cukur et al., 2013; Giallanza et al., 2023; Ralph et al., 2017).

Giallanza et al. (2023) provide an example of this using similarity data. They collected a dataset of 46 "round things," consisting of 21 balls used in sports (e.g., tennis ball, basketball) and 25 roughly spherical fruits/vegetables (e.g., orange, pumpkin). Each object was associated with a ground-truth average diameter, which ranged from 0.31 cm to 25.6 cm. Participants were asked to make similarity judgments in one of two conditions: "size" or "kind". In the size condition, participants were shown two options and one target and asked to choose "which option is more similar in size to the target". In the kind condition, participants instead chose "which option is a more similar kind of thing to the target". From these similarity judgments, Giallanza et al. (2023) computed 2-dimensional embeddings using Multi Dimensional Scaling (Shepard, 1962). They computed embeddings separately for each of the two conditions, demonstrating that the representations differ based on the task, with kind information represented more strongly in the kind condition and size information represented more strongly in the size condition.

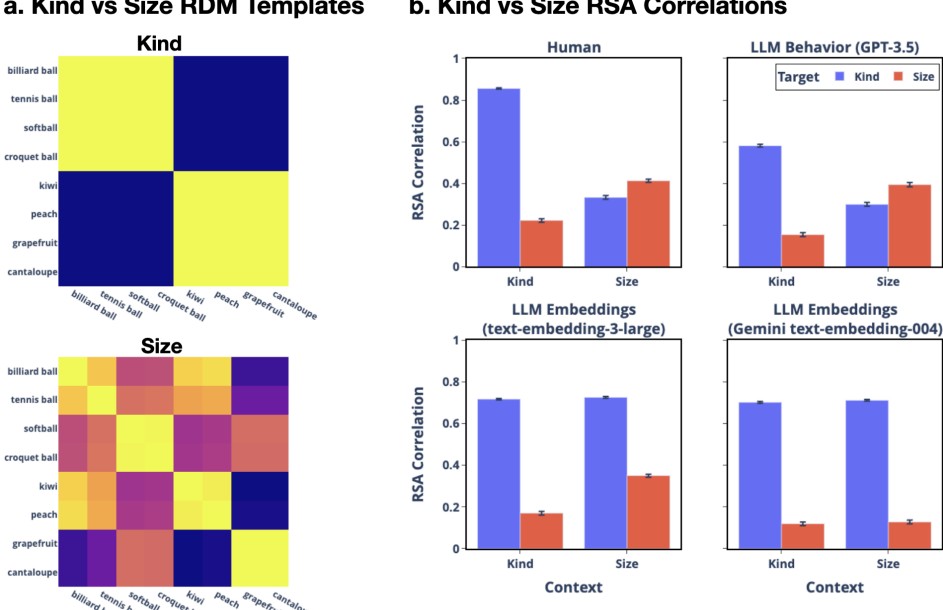

Figure 2: **Task context effects in humans & LLMs.** a) RDM templates for a subset of items in the kind and size contexts. Yellow squares represent similar object pairs, while blue squares represent dissimilar ones. b) Correlation between human/model representations and RDM templates. Blue bars indicate correlations with the kind RDM template, while red bars indicate correlations with the size RDM template.

### 3.1 RESULTS

We tested how well both LLM embedding representations and LLM behavior capture this form of context sensitivity using Representational Similarity Analysis (RSA; Kriegeskorte et al., 2008). RSA provides a measure of how well two representation spaces are aligned by calculating the correlation between the pairwise distances among items in each space. We used RSA to calculate how strongly LLM representations and behavior encode size vs kind information in each of the two experimental contexts. First, we generated a target pairwise distance matrix (Representational Dissimilarity Matrix, or RDM) for each of the two contexts (Figure 2a). For the kind task, the RDM contained a "0" if the two items belonged to the same category (fruit/vegetable vs ball) and a "1" if not. For the size task, the RDM contained the difference between the logarithms of the sizes of the ground-truth diameters of the two items (following Giallanza et al., 2023).

We compared the target RDMs to human-derived representations, LLM embedding representations and LLM behavior. For human representations, we used the 2-dimensional MDS embeddings from Giallanza et al. (2023), measuring the pairwise cosine distance between the object embeddings to create an object-by-object RDM for each of the two contexts. For LLM embedding representations, we calculated embeddings using two state-of-the-art text embedding models: text-embedding-3-large from OpenAI, and text-embedding-004 from Google Gemini. We calculated embeddings for each combination of object and task in Giallanza et al. (2023) (e.g., one embedding for basketball in the size task and a different embedding for basketball in the kind task) by providing one of two inputs to the embedding model: "What type of object is a X?" or "How big is a X?". We then calculated the pairwise cosine similarity between the representations in each of the tasks, resulting in one object-by-object RDM for each of the two tasks. For LLM behavior, we asked GPT-3.5 (gpt-3.5-turbo-0125) to make context-constrained pairwise similarity comparisons between the objects in the dataset: either in terms of kind (How similar are an apple and a baseball on a scale of 1-10?) or size (how similar are the sizes of an apple and a baseball on a scale of 1-10?). We used the pairwise similarity scores to construct an object-by-object RDM for each of the size and kind tasks.

We then tested how strongly the RDMs measured for human representations, LLM embedding representations, and LLM behavior represent context-relevant information. If the representations selectively encode context-relevant information, it should be the case that the RDM from the "kind" context has a greater correlation to the "kind" target RDM than to the "size" target RDM, and vice versa for the "size" context. Figure 2b depicts these correlations. The results indicate that both human representations and LLM behavior do selectively emphasize context-relevant information. However, neither of the two LLM text embedding models show the same degree of context sensitivity. Instead, both models represent kind information (blue bars in Figure 2b) more strongly than size information (red bars) in both contexts.

## 4 DISCUSSION

A full account of semantic cognition will explain not only how humans represent and access conceptual knowledge, but also how those representations and behaviors change in different contexts. In this article we evaluated whether LLMs can play a role in helping cognitive scientists answer these questions. We found that, like humans, LLMs adapt their semantic judgments to different contexts. This was true for both local context – similarity judgments for the same item pair varied based on which other pairs were present in the prompt – and task context – similarity judgments changed when reasoning about the kind versus the size of objects. In combination with prior work (e.g., Bhatia, 2023; Bhatia & Richie, 2022), these results suggest that the behavior of LLMs and humans are relatively well-aligned on a wide range of semantic decision making tasks. However, we also found that LLM text embeddings did not show strong context effects: The representations encoded category information more strongly than size information, even given a prompt that specifically asks about the size of the object. This reveals a potential limitation of LLM embeddings as models of human semantic representations, suggesting that LLM embeddings cannot be used as stand-ins for human representations in tasks that require context-dependent processing, such as tasks involving subordinate meanings (Hoffman et al., 2018), attending to uncommon features (Corbett et al., 2011), or suppressing strong associates (Mednick & Halpern, 1968).

The discrepancy between the behavior and the embedding representations of LLMs has two broader implications. First, it suggests that even in tasks where LLMs demonstrate human-like behavior, they may arrive at that behavior using representations very different from humans (e.g., in inductive reasoning tasks; Han et al., 2024). Second, it reveals a limitation with using similarity to probe representation structure. We showed that asking GPT-3.5 to make pairwise similarity judgments resulted in qualitatively different representations than measuring embedding representations from LLM text embedding models. This presents a potential challenge to the common assumption in cognitive science that pairwise similarity ratings from *humans* reflect distance in the psychological or neural space used to represent those items Tversky (1977), particularly in cases where those representations may be strongly influenced by context. In summary, while we have shown that LLMs and humans exhibit similar context sensitivities when making similarity judgments, we caution cognitive scientists against assuming that LLM embeddings reflect the dynamic nature of human conceptual knowledge.

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
