# OpenReview forum: "Context-Sensitive Semantic Reasoning in Large Language Models"
_ICLR.cc/2024/Workshop/Re-Align — ICLR 2024 Workshop Re-Align Poster_

### Official Review · Reviewer_ndVG · 2024-02-22
**Interesting and well-written paper but with some technical concerns**

**Rating:** 2
**Fit:** 3
**Confidence:** 3

**Workshop Review:**

### Summary
The authors compare LLM behavior and embeddings to humans on two semantic similarity judgment tasks manipulating local and task context. They find correspondence in context-sensitivity for several behavioral comparisons, but not for an embedding comparison.

### Clarity

Strengths:
- The paper is very clearly written.
- The background of the corresponding cognitive science literature is well-covered.
- The use of ample examples guides the reader.

Weaknesses:
- For the statistical tests reported in the discussion of Fig 1a, please include a 95% confidence interval for the effect, the precise statistical test used (e.g., one-sample two-tailed t-test) as well as the t-statistic and df instead of only including the p-value. Also, describe the "Context Effect" metric in more detail.
- In the discussion of the Giallanza task please include N participants.
- It is unclear how LLM behavioral judgments were derived for the second task. The authors describe what Qs were prompted, but not the space of outputs the LLM would be able to generate (e.g., rate on a scale of 1-10, etc.).

### Correctness

Strengths:
- It is great that the authors used the same prompts and experimental paradigm for the LLMs as to what the humans saw during the first behavioral experiment; this should be the norm, but is often not matched perfectly in many studies.
- The analysis in Figure 1B helps explain the GPT-3.5 data.
- In the comparison between the human and LLM behavior in the second experiment, using the same MDS procedure for both is great for comparing these spaces.

Weaknesses:
- It is not equivalent to use MDS to scale human and LLM behavioral judgments (which specifically has access to the dimension of interest) and then simply applying a linear PCA transformation to reduce the LLM embedding dimensionality to 2D (which is guided by maximizing overall variance, not variance in the dimension of interest). Seeing as the size feature does appear to be represented by the embeddings, just not in the first 2 PCs, it might be more helpful to characterize e.g., how many PCs are needed to decode size in the size-prompted condition vs in the kind-prompted condition (i.e., to see if the information becomes more readily available based on context despite being primarily absent from the first 2 PCs). The choice to only consider the first 2 PCs is somewhat arbitrary and driven by visualization (whereas it looks like this information comes online ~PC5 in Fig 3).
- The reliance on `text-embedding-3-small` uses a different/smaller LLM backend than `gpt-3.5-turbo-0125`. If the authors want to make a negative claim about the context sensitivity of LLM embeddings in general, they should evaluate the embeddings of the same model they generate the behavioral results from. For example, it is quite possible a larger model would have context-sensitive embeddings and this small model would not show the context-sensitive behavioral results. While this won't be possible from the OpenAI API, using an open-source LLM would allow this analysis. I think this is pretty crucial for making the core claim of the paper. Without it, this is a serious limitation that needs to be mentioned and the core claims dialed back.

### Novelty & Interest to the community

Strength:
- This is a fascinating line of inquiry and is of interest to members of the workshop.

**Reason For Not Giving Higher Score:**

I am concerned about the technical validity of the embedding results, which are not well controlled and could be explained by several other factors that are not addressed.

**Reason For Not Giving Lower Score:**

The core positive claims of the paper, e.g., the correspondence between LLM and human behavioral judgments are interesting and exciting. They will promote good discussion.

**Reviewer Domain:**

cognitive science

---

### Official Review · Reviewer_ktFT · 2024-02-25
**Great topic, conclusions may need revising**

**Rating:** 2
**Fit:** 3
**Confidence:** 2

**Workshop Review:**

This paper examines the effects of sentence context on LLM behavior (Study 1) and embeddings (Study 2), comparing them with previously reported results showing context sensitivity in humans. I have substantive methodological reservation regarding Study 2 and some presentational suggestions regarding Study 1.

Study 1:
- Could you provide more details about the way the model was prompted (provide the exact prompt, even if duplicated from Tversky & Gati)? How were the answers scored? If you provide all country pairs at the same time, does the model then spit out 8 numbers, and you match each number to the country pair in order?
- Depending on answers to the above: one reason why GPT 3.5 might not perform well on the task is the complexity of the prompt; might be worth mentioning
- In Figure 1, could you show the raw scores instead of just the difference? Lots of hidden information here. It would also be helpful to actually show the values for each country pair
- provide a conclusion to the study; the transition is a bit abrupt

Study 2:
I have 2 serious concern here: one, the use of MDS vs PCA; two, the claim that behavioral alignment and representational alignment are separable when the two are actually measured under different conditions.
- MDS is a non-linear projection technique, whereas PCA is linear. Initially, the authors use them to compare within-metric representations, which is perhaps somewhat justifiable (although ideally the dimensionality reduction method would be the same). But then they present the decoding analysis, which is completely unfair to LLM embeddings - precisely because their dimensionality reduction technique was linear, and the decoding probe is linear, so at no point are they allowed the nonlinear flexibility of MDS. So I think the conclusion that size and kind are more explicitly represented in similarity judgments than in LLM embeddings is flawed.
- The conclusion in the abstract is "behavioral alignment does not necessitate representational alignment". The main reason why I think this study cannot support this claim is because the tasks used to elicit the behavior and extract the representations are distinct! The prompts used are: "How similar are a X and a Y?" and “What type of object is a X?”, respectively, so any differences between representation and behavior might be due to that prompt difference. A much cleaner setup would be to use the same comparison prompt ("How similar are a X and a Y?"), get representations for Y given all the Xs, and analyze the representations of Y in the context of both the property (kind/size) and the comparison object (X). If the comparison object is not of interest, then representation of X rather than Y can be extracted. Either way, if the prompts are different, representational and behavioral alignment cannot be directly compared.

Other Study 2 comments:
- The idea behind Figure 2 is to show that similarity spaces are more context sensitive than embedding spaces. But the LLM behavior column is somewhat misleading - if one is to rotate and flip the bottom image (perfectly allowable transformations since directions do not have intrinsic meaning), the top and bottom images become quite similar
- Why use linear decodability (which is dependent on the size of the data, choice of the mapping model etc) rather than something more generic like RSA? RSA, in fact, is specifically developed to compare representational similarities, and thus seems like a perfect choice here. Linear decodability is a much more rough measure: e.g., size might be equally decodable in 2 systems but drive representations much more strongly in one of them.

**Reason For Not Giving Higher Score:**

Methodological concerns

**Reason For Not Giving Lower Score:**

Of interest to the field and the workshop audience

**Reviewer Domain:**

neuroscience

---

### Decision · Program_Chairs · 2024-03-02

Accept (Poster)